# Socioeconomic differences in participation and diagnostic yield within the Dutch national colorectal cancer screening programme with faecal immunochemical testing

**Miriam P. van der Meulen[1], Esther Toes-Zoutendijk[1]\*, Manon C. W. Spaander[2], Evelien Dekker[3], Johannes M. G. Bonfrer[4], Anneke J. van Vuuren[5], Ernst J. Kuipers[2], Folkert J. van Kemenade[5], M. F. van Velthuysen[5], Maarten G. J. Thomeer[6], Harriët van Veldhuizen[7], Harry J. de Koning[1], Iris Lansdorp-Vogelaar[1], Monique E. van Leerdam[8]**

1 Department of Public Health, Erasmus MC University Medical Center, Rotterdam, The Netherlands, 2 Department of Gastroenterology and Hepatology, Erasmus MC University Medical Center, Rotterdam, The Netherlands, 3 Department of Gastroenterology and Hepatology, Academic Medical Center, Academic University Medical Centers, Amsterdam, The Netherlands, 4 Department of Clinical Chemistry, Netherlands Cancer Institute–Antoni van Leeuwenhoek Hospital, Amsterdam, The Netherlands, 5 Department of Pathology, Erasmus MC University Medical Center, Rotterdam, The Netherlands, 6 Department of Radiology, Erasmus MC University Medical Center, Rotterdam, The Netherlands, 7 Department of Quality Improvement, Erasmus MC University Medical Center, Rotterdam, The Netherlands, 8 Department of Gastroenterology and Hepatology, Netherlands Cancer Institute–Antoni van Leeuwenhoek Hospital, Amsterdam, The Netherlands

\* e.toes-zoutendijk@erasmusmc.nl

**Data Availability Statement:** The authors note that the sharing of the speudonymised dataset is not

## Abstract

### Background

CRC mortality rates are higher for individuals with a lower socioeconomic status (SES). Screening could influence health inequalities. We therefore aimed to investigate SES differences in participation and diagnostic yield of FIT screening.

### Methods

All invitees in 2014 and 2015 in the Dutch national CRC screening programme were included in the analyses. We used area SES as a measure for SES and divided invitees into quintiles, with Quintile 1 being the highest SES. Logistic regression analysis was used to compare the participation rate, positivity rate, colonoscopy uptake, positive predictive value (PPV) and detection rate across the SES groups.

### Results

Participation to FIT screening was significantly lower for Quintile 5 (67.0%) compared to the other Quintiles (73.0% to 75.1%; adjusted OR quintile 5 versus quintile 1: 0.73, 95%CI: 0.72–0.74), as well as colonoscopy uptake after a positive FIT (adjusted OR 0.73, 95%CI: 0.69–0.77). The detection rate per FIT participant for advanced neoplasia gradually increased from 3.3% in Quintile 1 to 4.0% in Quintile 5 (adjusted OR 1.20%, 95%CI 1.16–

allowed. Researchers had access to the data for the purpose of evaluation of the screening programme. The data is owned by the samenwerking bevolkingsonderzoeken (FSB). All data of the national CRC screening programme is stored in the national database ScreenIT. Data of this study as well as other data of the national CRC screening programme are only available with the permission of FSB. Researchers interested in accessing and analysing data of the national CRC screening database ScreenIT may contact the data science officer of BVO-NL (wetenschappelijkonderzoek@fsb-ssc.nl). Data is obtained through the population based national screening programme. Permit for carrying out the national screening programme is laid down in the Population Screening Act. The researcher were not involved in data collection and had only access to pseudonymised individual level data By participating individuals gave their implied informed consent. All individuals could object to use their data for scientific research.

**Funding:** This analysis was performed as part of the national monitoring and evaluation of the Dutch colorectal cancer screening programme, funded by the National Institute of Public Health and Environment (Rijksinstituut voor Volksgezondheid en Milieu (RIVM)). The funders had no role in study design, data collection and analysis, decision to publish, or preparation of the manuscript.

**Competing interests:** The authors have declared that no competing interests exist.

1.24). As a result of lower participation, the yield per invitee was similar for Quintile 5 (2.04%) and Quintile 1 (2.00%), both being lower than Quintiles 2 to 4 (2.20%-2.28%).

## Conclusions

Screening has the potential to reduce health inequalities in CRC mortality, because of a higher detection in participants with a lower SES. However, in the Dutch screening programme, this is currently offset by the lower participation in this group.

## Introduction

Colorectal cancer (CRC) is the second most common cause of cancer-related mortality in the Western world [1]. Screening can prevent a considerable part of these deaths by early detection and treatment of CRC and its precursor lesions. Therefore, various countries and local initiatives across the world have adopted population-based screening for CRC [2, 3], aiming for equal access to CRC screening for the entire population. In Europe, CRC mortality rates are consistently shown to be higher among individuals with a lower SES (SES) [4]. Since screening can reduce CRC mortality and CRC incidence depending on screening methods and screening uptake, it has the potential to decrease these health inequalities.

However, if the participation to and performance of the screening programme differ across SES groups, screening may fail to reduce or even augment health inequalities. Indeed, several studies demonstrated that lower SES groups had lower participation rates in CRC screening with colonoscopy, guiac faecal occult blood test (gFOBT) and faecal immunochemical test (FIT) [5–11]. Less is known about the participation to subsequent colonoscopy and the performance of a screening programme across SES groups in terms of positivity rate and diagnostic yield. A large study using gFOBT showed that the most deprived individuals had a higher positivity rate and no difference in positive predictive value (PPV) [7]. As far as we know, only one small study from the Basque country using FIT showed a similar PPV among SES groups and a higher detection rate in deprived men (but not in women) [12].

Because many organized screening programmes across the world have chosen to use FIT [3], it is important to get more insight into the potential impact of a FIT screening programme on inequalities in health. Data from the Dutch national CRC screening programme with FIT enabled us to investigate SES differences in participation and diagnostic yield with FIT screening.

## Methods

### Dutch CRC screening programme

The Dutch national CRC screening programme using biennial FIT was introduced in 2014 with a gradual roll-out by age within a period of five years. The target population will eventually consist of individuals aged 55 to 75 years. The target population receives a pre-invitation letter by post, followed by an invitation letter by post together with a single FIT sampling device (FOB-Gold, Sentinel, Italy). As a result of the gradual roll-out, in 2014 only individuals aged 63, 65, 67, 75 and 76 years and in 2015 only individuals aged 61, 63, 65, 67, 69 and 75 years were invited. The first half year of 2014, the cut-off level for referral to colonoscopy was 15 μg Hb/g faeces, thereafter, the cut-off level was increased to 47 μg Hb/g faeces because of higher than expected participation rate, positivity rate, and a lower than expected PPV [13].

We present the data at a cut-off level of 47 μg Hb/g faeces, also for the individuals screened with the lower cut-off level. All data of the screening programme are continuously collected in a national information system of the CRC screening programme (ScreenIT). ScreenIT includes personal details (like sex, date of birth, place of residence, postal code), FIT results, medical details from the pre-colonoscopy intake and colonoscopy results from endoscopy centres and pathology diagnoses from the national pathology registry PALGA. The Dutch screening programme is described in more detail in a previous publication [13].

## Measuring socioeconomic status

Area SES, based on the postal code, was used as a measure for SES. The Dutch postal code consists of four-digits and two letters, of which the four-digit postal code of the invitees' place of residence was used. Scores per four-digit postal code were provided by The Netherlands Institute for Social Research [14]. The provided SES scores per postal code are calculated with a principal components analysis based on income, employment status and educational level [14]. Socioeconomic data of 2014 were used. The scores based on postal codes were divided into quintiles based on the rank of the scores, corrected for the number of individuals (of all ages) living in the postal code areas. The population in the quintiles was calculated with data on the number of inhabitants per age-group in each postal code in 2014 [15]. Quintile 1 was the quintile with highest SES, with the highest scores (high income, high employment rate, high educational level), while Quintile 5 was with the lowest SES.

## Background incidence

Background incidence of CRC across SES groups prior to the introduction of screening was determined as comparator for the yield in FIT participants. All CRC diagnoses from 2008 till 2012 were obtained from the Dutch Cancer Registry (NKR), with the year of diagnosis, the age of the patient at diagnosis and the SES. The SES was determined as described earlier but based on SES scores and population numbers in 2010.

## Analysis

**National screening programme.** Data on the invitees of 2014 and 2015 were collected until 31 March 2016. Outcomes were 1) participation rate of FIT screening, 2) positivity rate of FIT, 3) colonoscopy uptake after a positive FIT, 4) positive predictive value (PPV) for advanced neoplasia (AN, advanced adenomas and CRC combined) and CRC alone, 5) detection rates per participant and 6) yield per invitee of AN and of CRC.

The FIT participation rate was defined as the number of persons returning a stool sample divided by the number of persons invited. Positivity rate was defined as the number of participants with a test result at or above the cut-off level divided by the number of participants with an assessable stool sample. Cut-off level for a positive test result was 47 ug Hb/g faeces. Positive tests with a result between 15 and 47 ug Hb/g faeces of individuals screened with the lower cut-off level of 15 ug Hb/g faeces were considered as a negative test result and all data collected after the positive test, such as colonoscopy uptake and detected lesions, were not included. The colonoscopy uptake was defined as the number of persons who underwent a colonoscopy divided by the number of persons with a positive FIT.

The PPV of AN and CRC was calculated as the number of persons with AN or CRC respectively, divided by the number of persons who underwent a colonoscopy. An advanced adenoma was defined as any adenoma with 1) histology showing ≥25% villous component or 2) high-grade dysplasia or 3) size ≥10 mm. The detection rate (DR) was defined as the number of persons with AN and CRC detected during colonoscopy divided by the number of screened

persons with an assessable stool sample, (assuming full compliance to colonoscopy). Similarly, the yield per invitee was calculated as the number of persons with AN and CRC detected during colonoscopy divided by the number of invitees.

Proportions were determined by descriptive analyses. Logistic regression analysis was performed to estimate odds ratio (OR) of the quintiles on FIT participation rate, positivity rate, colonoscopy uptake, PPV for AN and for CRC and detection rate per invitee for AN and for CRC. In this multivariable regression, outcomes were corrected for well-known confounders: age (continuous) and sex. To determine the DR per FIT participant, we performed poststratification (including sex and age) to adjust for the differences in colonoscopy uptake across SES quintiles and assumed full compliance.

The analyses were conducted with R-3.2.3.

**Background incidence.** Age-standardised incidence rates were calculated by direct standardisation to the European Standard Population (Eurostat 2013) [16]. All rates are presented as European age-standardised rates (ESR per 100,000), with 95% confidence intervals (CI). The incidence rate ratio (IRR) was calculated by dividing the ESR of each SES quintile with the corresponding ESR of Quintile 1 (the highest socioeconomic quintile), 95% CI were determined.

## Sensitivity analysis

In the sensitivity analyses we replicated all analyses with SES divided in deciles instead of quintiles.

## Data availability and ethical approval

The authors note that the sharing of the pseudonymised dataset is not allowed. Researchers had access to the data for the purpose of evaluation of the screening programme. The data is owned by the BVO-NL. All data of the national CRC screening programme is stored in the national database ScreenIT. Data of this study as well as other data of the national CRC screening programme are only available with the permission of BVO-NL. Researchers interested in accessing and analysing data of the national CRC screening database ScreenIT may contact data officer of BVO-NL ((wetenschappelijkonderzoek@fsb-ssc.nl).

Data is obtained through the population based national screening programme. Permit for carrying out the national screening programme is laid down in the Population Screening Act. The researchers were not involved in data collection and had only access to pseudonymised individual level data. By participating individuals gave their implied informed consent. All individuals could object to use their data for scientific research.

## Results

### Descriptive national screening programme

In 2014 and 2015, 1,882,916 individuals were invited for first round FIT screening, of whom 1,866,060 (99.1%) had an area-based SES score. Quintile 3 contained the largest proportion of invitees (Table 1). Of the invitees with SES score, 49.3% were male, ranging from 48.1% in Quintile 5 to 49.8% in Quintile 2. The invitees of Quintile 5 had a median age of 66.8 years compared with 65.9 years in the total population.

### Participation and positivity rate

With Quintile 1 as reference, participation to FIT screening was higher in Quintile 2 and 3 (Quintile 1 73.9%, Quintile 2 and 3: 75.1% (Table 2), but lower in Quintile 4 and Quintile 5,

**Table 1. Descriptive of the number, age and gender distribution of the invitees in each quintile.** Quintile 1 least deprived, Quintile 5 most deprived.

| | Number | % | Gender | | | Age | |
| --- | --- | --- | --- | --- | --- | --- | --- |
| | | | Males | % | | median | |
| Quintile 1 | 334233 | 17.9% | 166013 | 49.7% | | 65.7 | |
| Quintile 2 | 381344 | 20.4% | 189929 | 49.8% | | 65.8 | |
| Quintile 3 | 403907 | 21.6% | 199777 | 49.5% | | 66.0 | |
| Quintile 4 | 388664 | 20.8% | 191341 | 49.2% | | 66.4 | |
| Quintile 5 | 357912 | 19.2% | 172222 | 48.1% | | 66.8 | |
| **Total** | 1866060 | 100.0% | 919282 | 49.3% | p<0.001 | 65.9 | p<0.001 |

with the lowest participation rate in Quintile 5 (67.0%). Multivariate analysis showed an OR of 0.73 (95% CI 0.72–0.74) for Quintile 5 compared with Quintile 1. The positivity rate was lowest in Quintile 1 (5.8%) and gradually increased with increasing Quintile. The positivity rate of Quintile 5 (7.1%) had an OR of 1.22 (95% CI 1.20–1.25) compared to Quintile 1. Colonoscopy uptake after a positive FIT showed a similar pattern as the participation to FIT screening, with the highest uptake in Quintile 2 (82.4%) and significantly lower uptake in Quintile 4 and 5 (80.0% and 75.8% respectively) compared to Quintile 1 (81.3%) (OR Quintile 5 versus Quintile 1: 0.73 95% CI 0.69–0.77). In all regression analysis, age and sex were significant variables (p < 0.001).

## Diagnostic yield

The PPV for AN was highest in Quintile 3 (58.4%) and lowest in Quintile 5 (56.1%). Multivariate analysis showed an OR of 1.06 (95%CI 1.01–1.12) for Quintile 3 compared with Quintile 1 and an OR of 0.98 (95%CI 0.93–1.03) for Quintile 5 compared with Quintile 1. The PPV for CRC was also highest in Quintile 3 (9.6%, adjusted OR compared to Quintile 1 1.03 (95% CI

**Table 2. The participation to FIT, positivity rate and colonoscopy uptake after a positive FIT in each quintile, with the univariate and multivariate odds ratio (OR) and 95% CI.**

| Quintile | N | Attendance to FIT | OR (univariate) | OR (multi-variate)* | 95% CI | | | |
| --- | --- | --- | --- | --- | --- | --- | --- | --- |
| Quintile 1 | 246858 | 73.9% | 1 | 1 | | | | p<0.001 |
| Quintile 2 | 286527 | 75.1% | 1.07 | 1.07 | 1.06 | - | 1.08 | |
| Quintile 3 | 303133 | 75.1% | 1.06 | 1.07 | 1.06 | - | 1.08 | |
| Quintile 4 | 283640 | 73.0% | 0.96 | 0.96 | 0.95 | - | 0.97 | |
| Quintile 5 | 239945 | 67.0% | 0.72 | 0.73 | 0.72 | - | 0.74 | |
| | N | Positivity rate | OR (univariate) | OR (multi-variate)* | 95% CI | | | |
| Quintile 1 | 14466 | 5.8% | 1 | 1 | | | | p<0.001 |
| Quintile 2 | 17726 | 6.2% | 1.06 | 1.05 | 1.03 | - | 1.08 | |
| Quintile 3 | 19235 | 6.3% | 1.09 | 1.08 | 1.06 | - | 1.10 | |
| Quintile 4 | 19037 | 6.7% | 1.16 | 1.15 | 1.12 | - | 1.17 | |
| Quintile 5 | 17145 | 7.1% | 1.24 | 1.22 | 1.20 | - | 1.25 | |
| | N | Attendance to diagnostic colonoscopy | OR (univariate) | OR (multi-variate)* | 95% CI | | | |
| Quintile 1 | 11768 | 81.3% | 1 | 1 | | | | p<0.001 |
| Quintile 2 | 14612 | 82.4% | 1.08 | 1.08 | 1.02 | - | 1.14 | |
| Quintile 3 | 15732 | 81.8% | 1.03 | 1.04 | 0.98 | - | 1.10 | |
| Quintile 4 | 15234 | 80.0% | 0.92 | 0.93 | 0.88 | - | 0.98 | |
| Quintile 5 | 12992 | 75.8% | 0.72 | 0.73 | 0.69 | - | 0.77 | |

* The multivariate OR is corrected for age and gender.

0.95–1.11)) and lowest in Quintile 4 (8.5%, adjusted OR compared to Quintile 1 0.90 (95% CI 0.82–0.97)) (Table 3). The DR for AN in FIT participants was lowest in Quintile 1 (2.71% uncorrected and 3.33% corrected) and gradually increased with higher quintile (Quintile 5: 3.04% uncorrected, 4.01% corrected; OR 1.20 (95% CI 1.16–1.24)) (Table 4). The DR for CRC in FIT participants varied between the quintiles and was significantly higher in Quintile 5 with 0.52% (OR 1.17 (95% CI 1.08–1.27)) compared to Quintile 1. The yield of AN and of CRC in invitees was similar for Quintile 1 and 5, but both Quintiles had significantly lower yield than Quintiles 2 to 4 (Table 4). In all regression analysis, age and sex were significant variables ($p < 0.001$).

### Background CRC incidence

In total, 65,130 incident cases of CRC were recorded from 2008 to 2012. The European age-standardized rate was very similar across SES quintiles, varying from 456 per 100,000 in Quintile 1 to 462 per 100,000 in Quintile 5 and was highest in Quintile 4 with 471 per 100, 000 (IRR of 1.03) (Table 5).

### Sensitivity analyses

Using deciles of SES rather than quintiles led to similar patterns in participation, detection and yield, albeit the difference between SES groups was more pronounced (S1 Appendix). For instance, participation to FIT screening was lowest in Decile 10 with 64.3% compared to 72.6% in Decile 1 (adjusted OR 0.69, 95%CI: 0.68–0.70). The detection rate per FIT participant for advanced neoplasia gradually increased from 3.2% in Decile 1 to 4.1% in Decile 10 (adjusted OR 1.28%, 95%CI 1.24–1.33).

### Discussion

Our study showed a significantly lower participation to FIT screening and subsequent colonoscopy in case of a positive FIT for individuals in the lowest SES group. The participation was stable for high and moderate SES but decreased for individuals with a low SES. The

**Table 3. The positive predictive value (PPV) of FIT for advanced neoplasia (AN) and colorectal cancer (CRC) in each SES quintile, with the univariate and multivariate odds ratio (OR) and 95% CI.**

| | N | PPV AN* | OR (univariate) | OR (multi-variate)** | 95% CI | | | |
|---|---|---|---|---|---|---|---|---|
| Quintile 1 | 6689 | 56.8% | 1 | 1 | | | | p<0.001 |
| Quintile 2 | 8388 | 57.4% | 1.02 | 1.02 | 0.97 | - | 1.07 | |
| Quintile 3 | 9191 | 58.4% | 1.07 | 1.06 | 1.01 | - | 1.12 | |
| Quintile 4 | 8872 | 58.2% | 1.06 | 1.06 | 1.01 | - | 1.11 | |
| Quintile 5 | 7295 | 56.1% | 0.97 | 0.98 | 0.93 | - | 1.03 | |
| | N | PPV CRC* | OR (univariate) | OR (multi-variate)** | 95% CI | | | |
| Quintile 1 | 1103 | 9.4% | 1 | 1 | | | | p<0.01 |
| Quintile 2 | 1376 | 9.4% | 1.01 | 1.00 | 0.92 | - | 1.09 | |
| Quintile 3 | 1516 | 9.6% | 1.03 | 1.03 | 0.95 | - | 1.11 | |
| Quintile 4 | 1301 | 8.5% | 0.90 | 0.90 | 0.82 | - | 0.97 | |
| Quintile 5 | 1165 | 9.0% | 0.95 | 0.94 | 0.86 | - | 1.02 | |

*An advanced adenoma was defined as any adenoma with histology showing ≥25% villous component or high-grade dysplasia or with size ≥10 mm. The PPV was calculated as the number of persons with an advanced adenoma or with a CRC (together called advanced neoplasia (AN) divided by the number of persons who underwent a colonoscopy after a positive FIT.

**The multivariate OR is corrected for age and gender.

**Table 4. The detection rate (DR) per 100 participants uncorrected and corrected for colonoscopy uptake and the yield per 100 invitees of advanced neoplasia (AN) and colorectal cancer (CRC) for each quintile, with the univariate and multivariate odds ratio (OR) and 95% CI.**

| | N | DR PER PARTICIPANT | | | | | | | YIELD PER INVITEE | | | | | | |
| --- | --- | --- | --- | --- | --- | --- | --- | --- | --- | --- | --- | --- | --- | --- |
| | | DR AN uncorrected* | DR AN corrected** | OR (univariate) | OR (multi-variate)*** | 95% CI | | | yield AN | OR (univariate) | OR (multi-variate)*** | 95% CI | | |
| Quintile 1 | 6689 | 2.71% | 3.33% | 1 | 1 | | | p<0.01 | 2.00% | 1 | 1 | | | p<0.01 |
| Quintile 2 | 8388 | 2.93% | 3.55% | 1.09 | 1.07 | 1.04 | - | 1.10 | 2.20% | 1.10 | 1.10 | 1.06 | - | 1.13 |
| Quintile 3 | 9191 | 3.03% | 3.70% | 1.13 | 1.12 | 1.09 | - | 1.15 | 2.28% | 1.14 | 1.14 | 1.10 | - | 1.17 |
| Quintile 4 | 8872 | 3.13% | 3.91% | 1.16 | 1.18 | 1.15 | - | 1.21 | 2.28% | 1.14 | 1.13 | 1.10 | - | 1.17 |
| Quintile 5 | 7295 | 3.04% | 4.01% | 1.12 | 1.21 | 1.18 | - | 1.24 | 2.04% | 1.01 | 1.00 | 0.97 | - | 1.04 |
| | N | DR CRC* | DR CRC** | OR (univariate) | OR (multi-variate)*** | 95% CI | | | yield CRC | OR (univariate) | OR (multi-variate)*** | 95% CI | | |
| Quintile 1 | 1103 | 0.45% | 0.55% | 1 | 1 | | | p<0.01 | 0.33% | 1 | 1 | | | p<0.01 |
| Quintile 2 | 1376 | 0.48% | 0.59% | 1.08 | 1.06 | 1.01 | - | 1.12 | 0.36% | 1.09 | 1.09 | 1.00 | - | 1.18 |
| Quintile 3 | 1516 | 0.50% | 0.61% | 1.12 | 1.12 | 1.07 | - | 1.17 | 0.38% | 1.14 | 1.13 | 1.04 | - | 1.22 |
| Quintile 4 | 1301 | 0.46% | 0.57% | 1.03 | 1.05 | 0.99 | - | 1.10 | 0.33% | 1.01 | 1.00 | 0.92 | - | 1.08 |
| Quintile 5 | 1165 | 0.49% | 0.64% | 1.09 | 1.17 | 1.12 | - | 1.23 | 0.33% | 0.99 | 0.97 | 0.89 | - | 1.05 |

*An advanced adenoma was defined as any adenoma with histology showing ≥25% villous component or high-grade dysplasia or with size ≥10 mm. The detection rate was defined as the number of persons with advanced adenomas or with CRC (together called advanced neoplasia (AN)) detected during colonoscopy divided by the number of screened persons with an assessable stool sample.

**The detection rate was corrected for the differences in colonoscopy uptake compared to Quintile 1.

***The multivariate OR is corrected for age and gender and in the analysis per participant we corrected the DR for non-compliance to colonoscopy using poststratification (assuming full compliance).

positivity rate and detection rate of AN gradually and significantly increased with decreasing SES, while the PPV of AN and CRC was quite stable across SES groups.

Even though the participation was lower in Quintile 5, the participation rate of 67.0% in this Quintile was still higher than the desired 65.0% participation rate recommended by the European Union (EU) guidelines for quality assurance [17]. In contrast, the uptake of colonoscopy after a positive FIT was lower than the accepted 85% by the European Union (EU) guidelines for quality assurance for all quintiles (range 82.4%-75.8%), and was lowest for individuals

**Table 5. The number of colorectal cancer cases recorded between 2008 and 2012 and the European age-standardized ratio across the Quintiles of socioeconomic status, and the incidence rate ratio (IRR) of the Quintile compared to the most affluent Quintile (Quintile 1).**

| Quintile | Incident cases | ESR | | 95%CI | | | | IRR |
| --- | --- | --- | --- | --- | --- | --- | --- | --- |
| 1 | 11,123 | 456 | ( | 448 | - | 465 | ) | |
| 2 | 12,827 | 467 | ( | 459 | - | 475 | ) | 1.02 |
| 3 | 13,804 | 466 | ( | 458 | - | 474 | ) | 1.02 |
| 4 | 14,197 | 471 | ( | 463 | - | 478 | ) | 1.03 |
| 5 | 13,179 | 462 | ( | 454 | - | 470 | ) | 1.01 |

with a low SES. It is known that the uptake of colonoscopy in case of a positive FIT is higher than registered in the national screening database because some participants opt to have their colonoscopies at centres outside the screening programme. However, we do not expect that individuals with lower SES are more likely to perform the colonoscopy outside the screening programme than those with higher SES and thus do not expect that the observed SES gradient is the result of underreporting.

The SES difference in uptake of colonoscopy can in theory result from a higher prevalence of comorbidity among individuals with lower SES, resulting in exclusion for colonoscopy before or at intake. Another explanation for the association between SES and uptake of colonoscopy is the fact that colonoscopy after a positive FIT is considered standard medical care and is therefore covered by insurance companies. All citizens have an obligatory co-payment for delivered care during a calendar year ranging between €350 and €850. Therefore, individuals might omit to undergo the procedure or postpone the procedure if this co-payment maximum has not been reached in a given year. This may influence individuals to delay or even forego colonoscopy in order to avoid co-payments, particularly in lower SES.

The positivity rate gradually increased with decreasing SES. Because the PPV of FIT was stable across the SES range, the increase in positivity rate can only be caused by an increase in both true positive (the detection rate) and false positive FIT results. More false positive tests in low SES groups compared to high SES imply that FIT specificity is lower in low SES groups. A possible explanation for the lower specificity could be more comorbidity or anticoagulant use [18–20].

The increased detection rate in participants with lower SES can either be caused by a higher FIT sensitivity in lower SES for the same reasons as described for specificity or a higher CRC incidence in lower SES. We did not find a difference in CRC incidence by SES quintile in the period 2008–2012 (i.e. before the start of the implementation of the national screening programme). However, this does not preclude a difference in CRC incidence in those that participate to FIT across SES quintiles. If in lower SES groups individuals with symptoms are more prone to attend screening than individuals without symptoms ("unhealthy screenee bias"), or individuals with an immigrant background are less prone to participate than native Dutch individuals who have a higher CRC incidence, background incidence in the lower SES participants (in contrast with invitees) could be higher than in those with higher SES. Since a previous study observed similar stage distribution of screen-detected CRC across SES quintiles, the first explanation seems unlikely [21]. However, differences in participation between native Dutch and ethnic minorities on the other hand have been previously reported [22].

A strength of our study is the large sample size and high data completion rate due to the fact that data from different sources were automatically collected in the national screening database Screen-IT, like data on diagnostic yield of the screening programme. Our study also has a limitation; we did not have the personal SES, but based our analysis on the four-digit postal code. These aggregated data on SES may provide an inaccurate representation of the true individual SES. The use of area SES may diffuse results, therefore the observed differences could be more pronounced if linked to personal SES. In theory, there could be a mix of socio-economic classes in the middle quintiles, but less in quintile 5. In that case the drop in participation might be due to the lack of diffusion in the lowest SES areas.

In other countries with an organised FOBT-based screening programme the smallest socio-economic difference in participation was 6% (66% for lowest socioeconomic quintile and 72% for highest socioeconomic quintile), while the largest difference was 24% (42% versus 66%) [5]. With 67.0% for Quintile 5 versus 75.1% for the middle Quintiles, the difference in participation between SES groups in the Netherlands is at the lower end of this range. The difference between SES groups is also comparable to the differences in the breast cancer screening programme in

the Netherlands (participation rate of 79% for the lowest socioeconomic quintile up to 87% in the highest socioeconomic quintile) [23]. The SES differences in yield could also be compared to two other studies. One of those studies used gFOBT instead of FIT and showed a higher positivity rate in higher SES, opposite to our findings and a lower PPV for higher SES while we found a stable PPV [7]. A smaller study from the Basque country using FIT was more similar to our results, it showed a similar PPV among SES groups and a higher detection rate in deprived men (but not in women) with an OR of 1.38 (95% CI 1.23–1.55) [12].

Screening is often argued to increase already existing health inequalities. Based on our data, this is not observed in the Netherlands. Because of the higher yield in lower SES, it even has the potential to decrease health inequalities, however, this is currently offset by the lower participation in lower SES. It is therefore important to know the reasons behind the lower uptake in lower socioeconomic classes. In theory, patient preferences might be different and therefore lead to more individuals not undergoing screening due to a well-informed choice. However, it is more plausible that the lower participation in lower SES is not based on well-informed decision-making, since we previously found that across all quintiles only 12% of non-participants made an informed choice not to participate [24].

It is difficult to find interventions that decrease the socioeconomic gap in CRC screening. Several interventions have been found to increase overall uptake, such as the involvement of the family doctor. However, most did not reduce the socioeconomic gap or their influence on the socioeconomic gap was not assessed. To date, only two interventions have been demonstrated to reduce the gap, namely targeting specific groups [25] and sending an enhanced reminder letter with a banner that reiterates the screening offer [26]. Especially involvement of the family doctor after a positive screening test would be a plausible candidate for decreasing the SES gap in follow-up colonoscopy uptake. However, to recommend this and other specific interventions, further research is needed, also on the underlying reason for non-participation across the socioeconomic groups and to regional and ethnical differences in participation. This research could further clarify how to target groups that are less compliant and/or more at risk for AN and ensure well-informed decision-making.

In conclusion, screening has the potential to reduce existing socioeconomic inequalities in CRC mortality, because of a higher yield in participants with lower SES. However, this higher yield is currently offset by the lower participation in this group. Further research is needed into this lower participation to ensure well-informed decision-making.

## Supporting information

**S1 Appendix.**
(PDF)

## Acknowledgments

The authors thank the other members of the Dutch national colorectal cancer screening working group: A. van der Beek, J.A. Otte, Tj. Wiersma, A.A.M. Masclee, D.L. Schipper, F.J. van Kemenade, E.J.R. de Graaf, W.M.U. van Grevenstein, M. Frasa, L.H.J. Jacobs and J. Stoker for their contribution to the current national screening programme and their critical review of the manuscript.

## Author Contributions

**Conceptualization:** Miriam P. van der Meulen, Esther Toes-Zoutendijk, Iris Lansdorp-Vogelaar, Monique E. van Leerdam.

**Data curation:** Miriam P. van der Meulen.

**Formal analysis:** Miriam P. van der Meulen, Esther Toes-Zoutendijk.

**Methodology:** Miriam P. van der Meulen, Esther Toes-Zoutendijk, Iris Lansdorp-Vogelaar, Monique E. van Leerdam.

**Project administration:** Miriam P. van der Meulen.

**Writing – original draft:** Miriam P. van der Meulen, Esther Toes-Zoutendijk, Iris Lansdorp-Vogelaar, Monique E. van Leerdam.

**Writing – review & editing:** Miriam P. van der Meulen, Esther Toes-Zoutendijk, Manon C. W. Spaander, Evelien Dekker, Johannes M. G. Bonfrer, Anneke J. van Vuuren, Ernst J. Kuipers, Folkert J. van Kemenade, M. F. van Velthuysen, Maarten G. J. Thomeer, Harriët van Veldhuizen, Harry J. de Koning, Iris Lansdorp-Vogelaar, Monique E. van Leerdam.

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
