## [Decision Letter · Decision Letter 0]

7 Sep 2021

PONE-D-21-14958

Socioeconomic differences in participation and diagnostic yield within the Dutch national colorectal cancer screening programme with faecal immunochemical testing

PLOS ONE

Dear Dr. Toes-Zoutendijk,

Thank you for submitting your manuscript to PLOS ONE. After careful consideration, we feel that it has merit but does not fully meet PLOS ONE’s publication criteria as it currently stands. Therefore, we invite you to submit a revised version of the manuscript that addresses the points raised during the review process.

Please respond to the comments made by the reviewers and revise your manuscipt accordingly.

We look forward to receiving your revised manuscript.

Kind regards,

Frank T Kolligs, MD

Academic Editor

PLOS ONE

Journal Requirements:

2. In the ethics statement in the manuscript methods and the online submission form, please state whether the authors were involved in data collection, or had access to any identifying patient information at any time. Please also provide additional details about what information patients were provided about the use of their data for scientific research, and how they were able to opt out.

6.  Please include your tables as part of your main manuscript and remove the individual files. Please note that supplementary tables (should remain/ be uploaded) as separate "supporting information" files.

Reviewers' comments:

Reviewer's Responses to Questions

**Comments to the Author**

1. Is the manuscript technically sound, and do the data support the conclusions?

Reviewer #1: Yes

Reviewer #2: Yes

Reviewer #3: Yes

2. Has the statistical analysis been performed appropriately and rigorously? 

Reviewer #1: I Don't Know

Reviewer #2: Yes

Reviewer #3: Yes

3. Have the authors made all data underlying the findings in their manuscript fully available?

Reviewer #1: Yes

Reviewer #2: Yes

Reviewer #3: Yes

4. Is the manuscript presented in an intelligible fashion and written in standard English?

Reviewer #1: Yes

Reviewer #2: Yes

Reviewer #3: Yes

5. Review Comments to the Author

Reviewer #1: Van der Meulen et al present a retrospective population based study assessing whether the average socioeconomic status of the region where someone lives is associated with colorectal cancer screening uptake and diagnostic yield. The authors find a lower rate of screening in those people living in neighbourhoods with the lowest socioeconomic status, however the detection rate per FIT test performed was highest in those people living in these lower socioeconomic regions.

The major limitation is that the socioeconomic status is defined by a person’s geographic region rather than patient level data. However, this is a common issue with these types of studies and the results are important. I am supportive of accepting a revised manuscript.

Major Issues:

1. Please include more details in the methods about how the multivariate models were constructed, assumptions used, and in the results include information about how other variables impacted screening use and yield.

Minor Issues:

2. The word “program” is spelled program in some places and programme in others.

3. The paragraph formatting in the introduction is odd with some paragraphs having 1-2 sentences. These paragraphs should be merged.

4. Under methods they state that ScreenIT collects gender. They likely mean sex and not gender data is captured.

5. It would likely translate better to describe quintile 5 as the lowest socioeconomic status rather than the most deprived. For quintile 1, I would describe this as the highest socioeconomic status. This is just an optional style consideration and not mandatory.

Reviewer #2: This is a clearly written paper focussing on an important aspect of colorectal cancer screening on an epidemiological level. The methods used are described in an understandable way, the presentation of the results in text and tables is clear and precise. Finally, the interpretation of the results in the discussion section is very comprehensible. Two aspects may be mentioned in this context. The limitation attributed to failing personal SES is critically addressed. Of utmost imporatnce is the statement that CRC screening has the potential to reduce socioeconomically driven health inequalities instead of augmenting them.

I do not have any major criticisms, just some minor comments and suggestions.

Introduction, page 5, 2nd sentence: „Sreening can prevent part of these deaths“: This is expressed in an very conservative way! I woulf favor: „Screening can prevent a considerable part of theses deaths“.

Methods, page 7, last line: „DR“: write: detection rate (DE).

Methods, page 8, data availability: „FSB“: explain abbreviation!

Results, page 9, Participation and positivity rate: „7,2%“ – in table 2: 7,1%

Table 4 was missing, only legend.

Reviewer #3: Important study, well designed and focussed on a difficult topic which has been underestimated so far. Socioeconomic differences in relation to colorectal screening programs should be much more adressed. The study gives insights how this problem can be explained and possibly may be managed.

6. PLOS authors have the option to publish the peer review history of their article (what does this mean?). If published, this will include your full peer review and any attached files.

Reviewer #1: No

Reviewer #2: No

Reviewer #3: No

---

## [Author Response · Author response to Decision Letter 0]

18 Jan 2022

Reviewer #1: 

Major Issue:

1. Please include more details in the methods about how the multivariate models were constructed, assumptions used, and in the results include information about how other variables impacted screening use and yield.

Response: We thank the reviewer for this suggestion for clarification. We added a sentence to the methods. As sex and gender is not the focus of the paper, we only added the p-value of these factors to the results: 

Page 8, second paragraph:

“Logistic regression analysis was performed to estimate odds ratio (OR) of the quintiles on FIT participation rate, positivity rate, colonoscopy uptake, PPV for AN and for CRC and detection rate per invitee for AN and for CRC. In this multivariable regression, outcomes were corrected for well-known confounders: age (continuous) and sex.“

Page 10:

“In all logistic regression analysis, age and sex were significant variables (p < 0.001).” 

Minor Issues:

2. The word “program” is spelled program in some places and programme in others.

3. The paragraph formatting in the introduction is odd with some paragraphs having 1-2 sentences. These paragraphs should be merged.

4. Under methods they state that ScreenIT collects gender. They likely mean sex and not gender data is captured.

5. It would likely translate better to describe quintile 5 as the lowest socioeconomic status rather than the most deprived. For quintile 1, I would describe this as the highest socioeconomic status. This is just an optional style consideration and not mandatory.

Response: We thank the reviewer for pointing out these 4 issues, we have processed all 4 throughout the document. 

Reviewer #2: 

1. Introduction, page 5, 2nd sentence: „Sreening can prevent part of these deaths“: This is expressed in an very conservative way! I woulf favor: „Screening can prevent a considerable part of theses deaths“.

2. Methods, page 7, last line: „DR“: write: detection rate (DE).

3. Methods, page 8, data availability: „FSB“: explain abbreviation!

Response: We thank the reviewer for promoting these 3 clarifications, we have processed these in the document at the location as mentioned by the reviewer. 

4. Results, page 9, Participation and positivity rate: „7,2%“ – in table 2: 7,1%

Response: We thank the reviewer for noticing this error. The correct result is in the table, 7,1%, we adjusted the text on page 9. 

5. Table 4 was missing, only legend.

Response: We apologize for the omission of Table 4. We have now carefully checked to include this Table.

---

## [Editor Report · Decision Letter 1]

3 Feb 2022

Socioeconomic differences in participation and diagnostic yield within the Dutch national colorectal cancer screening programme with faecal immunochemical testing

PONE-D-21-14958R1

Dear Dr. ToesZoutendijk,

We’re pleased to inform you that your manuscript has been judged scientifically suitable for publication and will be formally accepted for publication once it meets all outstanding technical requirements.

Kind regards,

Frank T Kolligs, MD

Academic Editor

PLOS ONE

---

## [Editor Report · Acceptance letter]

8 Feb 2022

PONE-D-21-14958R1 

Socioeconomic differences in participation and diagnostic yield within the Dutch national colorectal cancer screening programme with faecal immunochemical testing 

Dear Dr. Toes-Zoutendijk:

I'm pleased to inform you that your manuscript has been deemed suitable for publication in PLOS ONE. Congratulations! Your manuscript is now with our production department. 

Kind regards, 

on behalf of

Dr. Frank T Kolligs 

Academic Editor

PLOS ONE